# Long Non-Coding RNAs (lncRNAs) as Regulators of the PI3K/AKT/mTOR Pathway in Gastric Carcinoma

**DOI:** 10.3390/ijms24076294

**Published:** 2023-03-27

**Authors:** Ismael Riquelme, Pablo Pérez-Moreno, Bárbara Mora-Lagos, Carmen Ili, Priscilla Brebi, Juan Carlos Roa

**Affiliations:** 1Institute of Biomedical Sciences, Faculty of Health Sciences, Universidad Autónoma de Chile, Temuco 4810101, Chile; 2Millennium Institute on Immunology and Immunotherapy (MIII), Center for Cancer Prevention and Control (CECAN), Department of Pathology, School of Medicine, Pontificia Universidad Católica de Chile, Santiago 8380000, Chile; 3Millennium Institute on Immunology and Immunotherapy (MIII), Laboratory of Integrative Biology (LIBi), Center for Excellence in Translational Medicine—Scientific and Technological Bioresource Nucleus (CEMT-BIOREN), Universidad de La Frontera, Temuco 4810296, Chile

**Keywords:** gastric cancer (GC), long non-coding RNAs (lncRNAs), PI3K/AKT/mTOR signaling pathway

## Abstract

Gastric cancer (GC) represents ~10% of the global cancer-related deaths, increasingly affecting the younger population in active stages of life. The high mortality of GC is due to late diagnosis, the presence of metastasis and drug resistance development. Additionally, current clinical markers do not guide the patient management adequately, thereby new and more reliable biomarkers and therapeutic targets are still needed for this disease. RNA-seq technology has allowed the discovery of new types of RNA transcripts including long non-coding RNAs (lncRNAs), which are able to regulate the gene/protein expression of many signaling pathways (e.g., the PI3K/AKT/mTOR pathway) in cancer cells by diverse molecular mechanisms. In addition, these lncRNAs might also be proposed as promising diagnostic or prognostic biomarkers or as potential therapeutic targets in GC. This review describes important topics about some lncRNAs that have been described as regulators of the PI3K/AKT/mTOR signaling pathway, and hence, their potential oncogenic role in the development of this malignancy.

## 1. Introduction

Cancer is a leading cause of death, especially in people younger than 70 years old in most countries of the world (112 of 183), which is explained by the higher longevity of the population and changes in the prevalence and distribution of the main risk factors for cancer, including those associated with socioeconomic development [1,2].

Among the different cancer types, gastric cancer (GC) is a significant public health problem, constituting the fifth most frequently diagnosed cancer and the third most lethal malignancy worldwide [3]. This cancer represents ~10% of global cancer-related deaths, increasingly affecting the younger population in active stages of life [3,4].

In general, the main clinical reasons for the high mortality rates found in GC—and in other malignancies—are (1) the late diagnosis of tumors due to vague or non-specific symptoms, which results in few individuals that can benefit from surgery (commonly considered the best treatment) [5,6,7,8]; (2) the fast and aggressive progression of many cancer types that leads to metastasis formation, inducing failure of other distant organs [7,8,9]; and (3) the development of intrinsic or acquired drug resistance in primary and/or metastatic tumors, which reduce therapeutic response and increases the recurrence rates and death in patients [8,9,10,11,12]. Unfortunately, the development of these aggressive and chemoresistant phenotypes are complex and multifactorial phenomenon [13,14,15,16].

As GC and other cancers constitute heterogeneous diseases both histologically and genetically, patient outcome is difficult to predict using the classical morphologic criteria and non-exhaustive molecular analyses [17]. Fortunately, recent advances in high-throughput technologies have led to the discovery of new molecular subtypes and novel molecular markers in order to understand better how intracellular pathways can induce certain processes of tumor progression. A comprehensive knowledge of these processes is pivotal to defining new therapeutic strategies for patients and providing broader options to avoid cancer progression in the future.

One of the main cell signaling pathways involved in the development of malignant phenotypes in GC is the phosphatidylinositol-3-kinases (PI3K)/AKT/mammalian target of the rapamycin (mTOR) pathway, which plays a key role in survival, cell growth, regulation of transcription and translation, and tolerance to therapeutics drugs [18,19].

In the last decades, a new group of molecules named non-coding RNAs (ncRNAs) has been described to play an important role in the regulation of cell signaling pathways involved in several malignant processes [20]. These ncRNAs include microRNAs (miRNAs), small-nuclear RNAs (snRNA), small-nucleolar RNAs (snoRNAs), ribosomal RNAs (rRNAs), small-interfering RNAs (siRNAs), PIWI-interacting RNAs (piRNAs), and long non-coding RNAs (lncRNAs), all of them with different action mechanisms able to regulate gene expression within the cells in both homeostatic and pathological conditions [3].

This review will address those lncRNAs that could directly or indirectly influence the different malignant characteristics induced by the PI3K/AKT/mTOR pathway in GC models, in order to obtain an updated perspective on how these transcripts would regulate the progression and aggressiveness of this neoplasm.

## 2. The PI3K/AKT/mTOR Signaling Pathway

In normal cells, the role of the PI3K/AKT/mTOR signaling pathway is to control multiple cellular functions such as protein synthesis, nutrient intake, metabolism, cell growth, proliferation, etc., by coordinating two main protein complexes: mTOR complex 1 (mTORC1), which is composed of mTOR, mLST8/GßL, Deptor, Raptor, and PRAS40 proteins [21]; and mTOR complex 2 (mTORC2), formed by mTOR, mLST8/GßL, Deptor, Rictor, mSIN1, and Protor proteins [22,23].

Briefly, the receptors of tyrosine kinase (RTKs) (e.g., EGFR and ERBB2) can activate the subunits p110α/β/γ/δ of PI3K, which can subsequently induce the activation by phosphorylation of AKT (p-AKT) through PIP3 or PDK1 [24,25,26,27]. Conversely, PTEN (a tumor-suppressor phosphatase) can reversely convert PIP3 into PIP2 to avoid AKT activation and maintain a controlled function of this axis in cells [18,28,29]. When AKT is phosphorylated, this can inhibit the dimeric protein complex TSC1/TSC2, resulting in the subsequent activation by phosphorylation of mTORC1 (p-mTOR), which induces increased activity of other downstream regulators such as P70S6K1, eIF4E, and 4E-BP1 that have specific cellular roles primordially—but not exclusively—related to cell proliferation, cell growth, and cell cycle progression [27,30,31,32,33,34,35]. These processes also require other downstream effectors for P70S6K1, eIF4E, and 4E-BP1 [21,36,37]. On the other hand, mTORC2 can also induce the activation of several substrates, including AKT, PKCα, and SGK1. In fact, AKT activation requires dual action via PI3K and mTORC2 to induce cell proliferation, cell survival and cell migration, and control various other metabolic processes [21] (Figure 1).

In summary, the PI3K/AKT/mTOR pathway activation requires the function of mTORC1 and mTORC2 complexes to integrate extra- and intracellular signals in order to promote protein synthesis, cell metabolism, growth, cell proliferation, apoptosis evasion, migration, invasion, and other features [38,39,40]. The normal function of this axis can be disrupted by genetic and epigenetic alterations that, in general, are able to induce an increased pathway activity in abnormal cells, leading to several types of diseases, such as cancer [41].

## 3. The Role of the PI3K/AKT/mTOR Signaling Pathway in Cancer

A higher expression (at transcriptional and translational levels) and activity (at a post-translational level) of most members from the PI3K/AKT/mTOR pathway is commonly found in cancer cells—independently of the type of malignancy. The deregulated activity of PI3K, AKT, mTORC1/mTORC2, or their corresponding downstream substrates can be the result of genetic alterations, of which the most frequent are the overexpressing/activating mutations in genes encoding RTKs (e.g., *EGFR* and *ERBB2* genes), PI3K subunits (*PI3KCA*, *PI3KCB, PI3KCG,* and *PI3KCD* genes), or AKT isoforms (*AKT1, AKT2*, or *AKT3* genes); and deletions or inactivating mutations in tumor suppressor genes (e.g., *PTEN* gene) [22,42,43,44,45].

These abnormal events can occur in only one target (gene or protein) or in more targets within this pathway. For instance, some gastric cancer (GC) cases have shown a high transcriptional and translational expression of the PI3K/AKT/mTOR pathway, with a mild overexpression of *PIK3CA*, *PIK3CB*, *AKT1*, *MTOR*, *RPS6KB1*, *EIF4EBP1*, and *EIF4E* genes, along with a slightly decreased expression of *PTEN* and *TSC1* gene in GC tissues compared to adjacent non-tumoral tissues [18]. Immunohistochemistry (IHC) analyses showed that PI3K, AKT, p-AKT, p-mTOR, p-4E-BP1, P70S6K1, p-P70S6K1, eIF4E, and p-eIF4E proteins were significantly overexpressed in tumor tissues compared to adjacent non-tumoral tissues (*p* < 0.05 for all) [18,29]. In fact, P70S6K1 expression was associated with advanced gastric tumors (pT3-pT4 tumors; *p* = 0.03); meanwhile, the levels of PI3K, AKT, p-AKT, P70S6K1, p-P70S6K1, and eIF4E were higher in tumors with lymph node metastases (*p* ≤ 0.01 for all) [29]. Conversely, PTEN was repressed in advanced GC (pT3-pT4) and in tumors with lymph node metastases [29]. In addition, many studies have shown that, in several cancer types, cell lines and tumor tissues often share those abnormal characteristics [18,46].

In addition, epigenetic alterations can also induce deregulation in this pathway through changes in gene methylation patterns, histone modifications, or deregulation in the expression of microRNAs (miRNAs) and long non-coding RNAs (lncRNAs) [41]. Both genetic and epigenetic alterations lead to an increased expression/activation of oncoproteins or reduced expression/activation of tumor-suppressor proteins. Therefore, in cancer cells, all the previously mentioned functions of the PI3K/AKT/mTOR pathway are found generally uncontrolled and increased, leading to the development of pro-tumorigenic features, including greater cell proliferation, cell survival, migration, invasion, and chemoresistance, among others [22,27,35,47,48].

## 4. Long Non-Coding RNAs (lncRNAs)

LncRNAs are defined as RNA sequences of more than 200 nucleotides, commonly transcribed by RNA polymerase II, and that typically do not possess functional open reading frames (ORFs) [49]. These transcripts act by regulating epigenetically the gene expression at post-transcriptional, transcriptional, translational, and post-translational levels by forming structures of RNA:RNA, RNA:DNA, and RNA:protein that allows them to participate in different cellular processes [50,51,52] (Figure 2).

In cancer, the abnormal expression of certain lncRNAs has been frequently associated with characteristics of aggressiveness in several types of malignancies, such as higher tumorigenic capacity, higher metastatic capacity, induction of epithelial-mesenchymal-transition (EMT) features, drug resistance, and stemness, all of them directly related to poor prognosis in cancer patients [53,54,55,56,57,58]. In GC, various lncRNAs have shown to promote the development of malignant features in cancer cells. Some of these lncRNAs could induce a direct or indirect interaction with genes, transcripts or proteins belonging to the PI3K/AKT/mTOR pathway, which usually correlates with a worsening of patients’ clinicopathological features.

## 5. The Clinical Significance of the PI3K/AKT/mTOR Pathway-Related lncRNAs in GC

Several lncRNAs related to the activation of the PI3K/AKT/mTOR signaling pathway have also been associated with various clinicopathological characteristics present in gastric tumors or prognostic indicators for subjects with this neoplasm (Table 1).

For instance, the high expression of ANRIL, CRNDE, UCA1, XLOC_006753, HAGLROS, NORAD, LOC101928316, AC093818.1, TMPO-AS1 and CCAT2 was associated with advanced TNM stage in GC tumors [59,61,62,64,65,66,67,69,70,71]; meanwhile, the overexpression of ANRIL, CRNDE, XLOC_006753, HAGLROS, AC093818.1, TMPO-AS1, CCAT2 and SNHG6 was correlated with poor survival of GC patients [59,61,64,65,69,70,71,74]. In addition, the high expression of ANRIL, XLOC_006753, NORAD, and CCAT2 was positively associated with tumor size [59,64,66,71], and the upregulation of CRNDE, HAGLROS, and AC093818.1 were correlated with a greater invasion depth of gastric tumors [61,65,69].

Overexpression of CRNDE, UCA1, XLOC_006753, NORAD, AC093818.1, TMPO-AS1, CCAT2, and MALAT1 has been correlated with lymph node metastasis or the presence of other types of metastases in GC patients [61,62,64,66,69,70,71,72]. In particular, AC093818.1 upregulation was associated with greater invasion, distal metastasis, being suggested as a highly sensitive and specific marker for GC metastasis [69]. More interestingly, higher levels of PVT1 and HIT000218960 have been associated with patients whose gastric tumors evidenced a certain degree of drug resistance to CDDP and 5-FU, respectively [60,73].

Regarding downregulated lncRNAs, GAS5 and PCAT18 have been inversely correlated with advanced TNM stage and greater tumor size, respectively [63,68]. Meanwhile, LOC101928316 repression has been associated with a higher differentiation degree and advanced TNM stage in GC cases [67].

All these results confirm the usefulness of lncRNAs as markers of prognosis, survival, and potential chemoresistance in GC that deserve further investigation. Furthermore, these data reaffirm the preponderant role of the PI3K/AKT/MTOR pathway in the first steps of gastric carcinogenesis and in the development of more severe phenotypes of this disease.

## 6. LncRNAs Involved in the PI3K/AKT/mTOR Pathway Activation in GC

Several lncRNAs have been shown to have any type of interaction with the PI3K/AKT/mTOR axis and could induce some malignant phenotypic features (Table 2).

For instance, CRNDE overexpression was shown to promote cell proliferation, migration, and invasion in MGC-803 and MNK-45 cells, by potentially increasing the expression of p-PI3K and p-AKT [61]. Similarly, the upregulation of UCA1 has also been shown to induce tumorigenic phenotype in BGC-823 cells and nude mice due to its potential effect on the higher protein expression of AKT3, p-AKT3, p-mTOR, and P70S6K1, and the inhibition of eIF4E expression in BGC-823 cells [62]. However, the reasons why EIF4E could be reduced in cells—even though it is a known activator of both mRNA translation and cell proliferation—were not further studied nor discussed by the authors.

In the case of XLOC_006753, its high expression was shown to promote cell proliferation, cell viability, and G1/S cell cycle progression in SGC-7901 cells by inducing higher expression levels of PI3K, p-AKT, p-mTOR, P70S6K1, p-P70S6K1, and p-4E-BP1 [64]. Likewise, CCAT2 promoted cell viability, colony formation, migration, invasion, and G0/G1 progression, and inhibited apoptosis and autophagy, by inducing a greater expression of p-mTOR, p-AKT, and p-P70S6K1 in cell lines HGC-27 and SGC-7901 [71]. Similar results were found for OGFRP1, which was shown to reduce apoptosis and promote proliferation, cell cycle progression, migration, and EMT process in human GC cells, as well as the development of greater tumors in nude mice models. All these effects are explained by the increased expression levels of p-AKT and p-mTOR caused by OGFRP1 [77].

Other lncRNAs have been mainly related to the role of autophagy in carcinogenesis. Examples of this are MALAT1, LIT3527, and SNHG6. MALAT1 is able to block autophagic flux in GC cells to subsequently induce the release inflammation mediators that would end in the development of a worse phenotype in GC. This effect can be generated by the inhibition that MALAT1 would cause on PTEN mRNA expression, thereby triggering a greater activation of its downstream members of the PI3K/Akt/mTOR pathway, such as AKT (p-AKT), mTOR (p-mTOR), and P70S6K1 (p-P70S6K1) [78]. Regarding LIT3527, this could inhibit apoptosis and autophagy, and induce greater migration in GC cells as well as promote lung metastasis in animal models by somehow increasing the levels of AKT, ERK, p-mTOR, and p-4EBP1 [79]. In the case of SNHG6, this lncRNA is able to increase cell viability and invasion and reduce apoptosis and autophagy by upregulating the levels of PI3K, p-PI3K, AKT, p-AKT, p-mTOR, and mTOR, and downregulating the levels of Beclin1 and LC3 [74].

Despite most articles not analyzing nor discussing the possible molecular processes by which these lncRNAs could regulate target genes/proteins within the PI3K/AKT/mTOR pathway, there are certain studies that do gather more information about the molecular function that these lncRNAs would have within cells. An example of this is AC093818.1, which promotes migration and invasion on in vitro and in vivo GC models, probably by binding to transcription factors STAT3 and SP1, and increasing the expression of PDK1, p-AKT1, and p-mTOR within GC cells. In other words, AC093818.1 seems to epigenetically induce an increased PDK1 expression in those GC models used in this study, which is concordant with the fact that PDK1 is an upstream inductor of the PI3K/AKT/mTOR pathway activation [69]. Another example is FOXD1-AS1, which accelerates the processes of tumor growth, metastasis, and chemoresistance on in vitro and in vivo GC models through two mechanisms: (1) its effects on miR-466 function (explained later) and (2) the fact that FOXD1-AS1 is able to promote the phosphorylation of 4E-BP1, and thereby, provokes the activation of eIF4E (p-eIF4E), which facilitates the translation of FOXD1 that finally is the one that will trigger these malignant features [80].

On the other hand, among downregulated lncRNAs, LOC101928316 has been shown to induce higher GC cell proliferation, migration, and invasion in vitro, and greater tumor weight in vivo by promoting in some way the higher expression of AKT3, mTOR, and p-mTOR, and a lower expression of PTEN. Therefore, LOC101928316 may be suggested as a tumor suppressor lncRNA [67].

## 7. LncRNAs That Sponge miRNAs to Activate the PI3K/AKT/mTOR Pathway in GC

Several lncRNAs act as ceRNAs (competing endogenous RNAs) by the formation of lncRNA-miRNA complementary binding, which results in miRNA sequestration in the cell and in abnormal expression of specific mRNAs that initially were silenced by those miRNAs [65]. Therefore, overexpression of a specific lncRNAs (ceRNAs) leads to lower levels of specific miRNAs, thereby increasing the expression of the corresponding target mRNA and its protein levels in the cell. Conversely, when a lncRNA (ceRNA) is downregulated occurs an increase in the levels of specific miRNAs that triggers the decreased expression of target mRNAs and proteins [80].

This ceRNA-involving post-transcriptional regulation has been also described, at different levels, in studies about the PI3K/AKT/mTOR pathway. For example, NEAT1 is a lncRNA highly expressed in GC tissues and cell lines that acts by absorbing miR-1294, thus inducing overexpression of the target of this miRNA, the mRNA of the *AKT1* gene. The higher AKT1 mRNA levels the higher AKT protein expression, which is a well-known inductor of cell proliferation and metastasis, and inhibitor of apoptosis in cells [77]. In the case of NORAD, its interaction with miR-214 could finally provoke a significant increase of p-AKT and p-mTOR levels in the cells to trigger a greater proliferation and other tumorigenic characteristics in both in vivo and in vitro GC models, as well as a reduction of apoptosis in GC cells [66].

On the other hand, HAGLROS overexpression is able to induce a higher expression of *MTOR* and other genes related with autophagy (e.g., *ATG9A* and *ATG9B*) through two manners: (1) HAGLROS can competitively bind miR-100-5p to avoid miR-100-5p-mediated mTOR mRNA inhibition, resulting in the increase of mTOR expression, and (2) HAGLROS interacts with the mTORC1 components to activate this complex and negatively regulate the autophagy signal in cells. This inhibition could be inducing the excessive proliferation and migration observed in vitro, and the development of tumorigenic features in vivo [65]. Similarly, the overexpression of FOXD1-AS1 has been described as accelerating the processes of tumor growth, metastasis, and chemoresistance on in vitro and in vivo GC models through two mechanisms: (1) the induction of FOXD1 translation (previously explained) and (2) the role of FOXD1-AS1 as a sponge of miR-466 in GC cells, thereby producing an increase in the expression of one of the targets of miR-466: the *PIK3CA* gene. The overexpression of p110α protein—encoded by *PIK3CA*—could also be responsible for the development of tumorigenic phenotypes [80].

Other lncRNAs can exert activation of the PI3K/AKT/mTOR pathway in an indirect manner, due to the corresponding competing miRNA has targets that are not canonical members of this signaling pathway. This is the case with TMPO-AS1, ANRIL, and MALAT1. LncRNA TMPO-AS1 was shown to promote cell proliferation and migration in GC cell lines, and angiogenesis in HUVEC cells by sponging miR-126-5p, which is a miRNA capable to induce transcriptional repression of the *BRCC3* gene. When the miR-126-5p expression is reduced in the cell, BRCC3 increases its expression and finally—in a manner not deeply explained yet—promotes a higher expression of p-PI3K, p-Akt, and p-mTOR proteins [70]. On the other hand, ANRIL has been also shown to induce epigenetic repression of miR-99a and miR-449a, which produces a higher expression mTOR and CDK6/E2F1 pathway by binding to PRC2, thus forming a positive feedback loop that continues promoting GC cell proliferation [59].

Another example of indirect activation on the PI3K/AKT/mTOR pathway via miRNA silencing is lncRNA MALAT1. In fact, Fu et al. indicated MALAT1 has another way to induce tumorigenic features in GC, which involves CCL21. Upregulation of CCL21 increases the MALAT1 expression to subsequently inhibit miR-202-3p expression, resulting in the upregulation of SRSF1. Then, SRSF1 is able to induce the expression of p-mTOR in order to promote migration, invasion, and EMT in vitro and greater tumor size and lower apoptosis in vivo [72]. Similarly, high LEF1-AS1 levels promote autophagy and inhibit apoptosis in GC cells due to its binding with miR-5100, which induces a higher expression of oncogene DEK and finally an increase in the expression of p-mTOR [81].

Unfortunately, for the cases of TMPO-AS1, ANRIL, MALAT1, and LEF1-AS1, the relationship between the respective BRCC3, PRC2, SRSF1, and DEK proteins and the ways that they induce upregulation of certain members within the PI3K/AKT/mTOR pathway are not deeply described/proposed in the literature.

As previously explained, when a lncRNA (ceRNA) is downregulated, an increase occurs in the levels of specific miRNAs, which triggers a reduced expression of target mRNAs and proteins. This is the case with GAS5, which can directly suppress the miR-222 function and, in this manner, avoid the successive interactions that could be produced by members of the PI3K/AKT/mTOR pathway in normal gastric cells. However, in GC cells, GAS5 is downregulated and miR-222 appears to be upregulated to induce lower PTEN protein levels, and subsequently, provoke activation of AKT and mTOR proteins, resulting in greater cell proliferation [63]. In a similar manner, GAS5 sponges miR-106a-5p to finally induce higher levels of p-AKT and p-mTOR in GC cells, which results in a rise of cell proliferation, migration, and invasion, and decline of apoptosis on in vitro models and greater tumor growth in vivo [75].

Another repressed lncRNA in GC is PCAT18, whose downregulation was shown to induce greater cell proliferation and cell cycle progression, and lower apoptosis in vitro, as well as increased tumor growth in vivo. Interestingly, the interactions PCAT18/miR-107/PTEN could be important to explain the high levels of p-PI3K found in GC cells when PCAT18 seems to be repressed [68]. Therefore, both GAS5 and PCAT18 may also be proposed as potential tumor suppressor lncRNAs in GC and could be explored to be used in GC treatment.

Although the complementary function between lncRNAs (ceRNAs) and miRNAs in oncogenesis is still not fully understood, several studies have confirmed that these interactions constitute potent mechanisms for post-transcriptional regulation in cancer [73]. Moreover, the potential interaction between lncRNAs and other types of non-coding RNAs (e.g., piRNAs, circRNAs, snoRNAs, etc.) cannot be dismissed and deserve to be more deeply analyzed.

## 8. LncRNAs Involved in Drug Resistance via the PI3K/AKT/mTOR Pathway in GC

As previously mentioned, the PI3K/AKT/mTOR pathway deregulation is also closely related to resistance to standard therapies in several types of neoplasms. Interestingly, activating mutations within this pathway are also considered one of the most important causes for intrinsic resistance in cancer cells [82]. Although many inhibitory drugs against targets of this pathway have been developed and these inhibitors—in combination with conventional chemotherapy—have mildly improved the overall survival rate in GC, drug resistance is still one of the main obstacles to curing this malignancy [3]. Therefore, it is essential to study other mechanisms that may be causing the resistant phenotype and other molecules that could help in treatment. In both cases, lncRNAs appear to be molecules of interest to study further.

Some lncRNAs have been associated to the activation of the PI3K/AKT/mTOR pathway and cisplatin (CDDP) resistance in GC. For instance, the upregulation of PVT1 was shown to be associated with higher CDDP resistance in GC cell lines by increasing *MTOR* mRNA levels, which at least induces apoptosis inhibition. In fact, PVT1 knockdown was able to reduce resistance in CDDP-resistant GC cell lines [60]. Similarly, FOXD1-AS1 has also been described as a promoter of CDDP resistance in MKN28 and BGC-823 cells and in mice with CDDP treatment due to its effect on the increased levels of p-4E-BP1 and higher expression of FOXD1 [80].

Another interesting example is LOC101928316, which is downregulated in CDDP-resistant AGS and BGC-823 cell lines. Interestingly, HDAC3 overexpression in resistant cells can reduce the transcription of LOC101928316 by inhibiting the acetylation of H3K4 on the LOC101928316 promoter. These alterations promote higher cell activity, cell invasion, and migration, but lower apoptosis in CDDP-resistant GC cells due to the effect of LOC101928316 on the increment in the levels of p-PI3K, p-Akt, and p-mTOR [76]. Therefore, all the previous studies suggest that PVT1, FOXD1-AS1, and LOC101928316 could serve as targets of CDDP resistance in GC.

Other lncRNAs have also been associated with the activation of the PI3K/AKT/mTOR pathway, but this time in relation to resistance to 5-fluorouracil (5-FU) in GC. This is the case of HIT000218960, which can increase the resistance to 5-FU in SNU-5 and NCI-N87 cells by upregulating the expression of HMGA2, resulting in higher phosphorylation of AKT, mTOR, and P70S6K1 [73].

On the other hand, XLOC_006753 has been shown to induce resistance to both 5-FU and CDDP in SGC-7901 cells. Research has shown that this resistant phenotype could be the result of the higher expression levels of PI3K, p-AKT, p-mTOR, P70S6K1, p-P70S6K1, and p-4E-BP1 that were induced by XLOC_00675 itself [60].

Undoubtedly, the research about the mechanisms by which lncRNAs are associated with resistance becomes an interesting field of study that requires further depth. The capacity that lncRNAs have shown to restore sensitivity to standard chemotherapy constitutes an attractive strategy for overcoming resistance during cancer treatments in the future because lncRNA-involving approaches could be used in those patients that evidence resistance to conventional drugs or could eventually develop resistance to chemical inhibitors and/or specific antibodies of the PI3K/AKT/mTOR axis.

## 9. LncRNAs as Potential Diagnostic, Prognostic, and Therapeutic Markers in GC

The literature has shown that certain lncRNAs (e.g., HULC, H19, LINC00152, and others) can be useful in the diagnosis or prognosis of GC due to their capability to be detected in tissues and plasma samples, also being associated with tumor progression and the clinical status of cancer patients [83]. For instance, this review previously described SNHG6 as a lncRNA with the potential to be a promising prognostic marker in this malignancy, given its correlation with the survival of GC patients [74]. Beyond the detection and quantification of lncRNA expression in clinical samples, the application of lncRNAs as biomarkers in daily practice could be challenging given their nature as RNA molecules. The ideal markers should be stable and easily detectable in plasma or other body fluids to allow non-invasive diagnosis; therefore, the analyses of lncRNAs contained in exosomes, microvesicles, apoptotic bodies, and circulating tumor cells must be encouraged over those analyses using free-circulating lncRNAs, which are more exposed to degradation made by serum RNAses [84,85,86]. In addition, lncRNAs in blood have emerged as a good form to improve personalized treatment selection and perform the follow-up of GC patients [86].

In terms of therapy, future studies must focus more on understanding the molecular and functional implications of specific GC-related lncRNAs, particularly their regulation on DNA, mRNA, miRNA, or protein activity. This knowledge might be useful to develop drugs or design RNA-based therapeutic strategies, such as, small-interfering RNAs (siRNAs), antisense oligonucleotides (ASOs), aptamers, synthetic mRNAs, or microRNAs (miRNAs) [87] against oncologic lncRNAs. RNA-based therapy has not yet been widely used in clinical trials, but it can offer some advantages compared to chemotherapy, small-molecule inhibitors, or antibodies: (1) its greater gene specificity within target cells, which avoids off-target binding and reduces toxicity found in other treatments; and (2) its capacity to pharmacoevolve their sequence at the same pace as disease [87]. These attributes give RNA-based therapeutics considerable potential to treat undruggable human diseases [87].

Genome editing by CRISPR-Cas9 or its variation CRISPR interference (CRISPRi) might also be considered a possible solution for some specificity limitations on lncRNA modulation and the subsequent phenotypical changes in cancer cells [52]; however, at present, there are no CRISPR-based developments clinically validated in order to modify lncRNA expression.

Despite RNA-based therapeutics and CRISPR-based methodologies having considerable potential to treat human diseases, these strategies cannot yet be used clinically due to two main problems: the delivery problems of these molecules into tumor cells and the toxicity that their chemical vehicles develop in patients [88,89].

## 10. Experimental and Technical Considerations for lncRNA-Involving Studies

The best manner to obtain reliable and reproducible results through the RNA-seq platform is by following certain basic requirements regarding a few parameters: (1) the experimental design, including the sample size of the study; (2) the suitability and quality of the samples analyzed, and (3) the type of non-coding RNA that is evaluated in the study.

Among the parameters of a good experimental design, it is essential to have an acceptable sample size between the groups to be contrasted in order to strengthen the comparative statistical analyses that allow a reliable discovery of differentially expressed RNAs. Working with the largest sample size possible will allow for gaining greater statistical power in the research results and will make results comparable with those from other robust studies.

Among the parameters of suitability and quality of samples analyzed, those studies using RNA-seq should consider evaluating only samples undergoing the best possible preservation protocols. In the case of lncRNAs, the best tissue samples to be used in RNA studies are fresh frozen tissues, because is the sample preservation technique that most likely prevents the degradation of RNA molecules by the action of nucleases performed at room temperature. Therefore, the protocols for collecting, biomolecule extraction, and pre-analytical procedures of fresh frozen tissues should be closely controlled to guarantee reliable, comparable, and accurate results. Sample quality is even more important when working with lncRNAs, which could be more susceptible to degradation because of their length. This degradation could induce falsely lower lncRNA expression levels or falsely higher expression of some shorter RNA transcripts, derived from degraded lncRNA sequences. When authors report the RNA quality indexes obtained before and after library preparation, such as RNA integrity number (RIN) values, A260/A280 ratios, and/or A260/A230 ratios, the study results become more reliable in the scientific community [90]. Unfortunately, few authors and article reviewers consider this information useful when results are published. More importantly, similar considerations should be encouraged in future studies assessing either free-circulating or exosomal lncRNA levels in serum samples, because RNA transcripts are usually exposed to harsh conditions in circulation (e.g., enzymes), which could decrease the stability of RNAs to subsequently perform RNA-seq.

As well as RNA-Seq, other methodologies have been developed to analyze the interactions induced by lncRNAs in order to elucidate their molecular roles either in physiological or pathological processes. For instance, some RNA pulldown approaches, such as RNA antisense purification (RAP), chromatin isolation by RNA purification (ChIRP), and capture hybridization analysis of RNA targets (CHART) methods, can provide information about the interactions between lncRNAs and DNA, protein, or other RNAs (e.g., mRNA, miRNAs, etc.), depending on the focus in which these experiments are used. These techniques can be complemented with Western blots, qPCRs, mass spectrometry, and high-throughput sequencing [91]. Alternatively, RNA immunoprecipitation sequencing (RIP-Seq) and photoactivatable ribonucleoside-enhanced crosslinking and immunoprecipitation (PAR-CLIP) complement the study of lncRNA–protein interactions [92]. All these techniques have been used more and more for exploring the functional mechanisms by which lncRNAs can regulate gene or protein expression to induce certain phenotypes in cancer cells. In this regard, most of articles cited in this review have focused on the discovery and validation of differentially expressed lncRNAs in GC, but only a few studies have deepened the functional evaluation of these transcripts.

Finally, another important issue to be considered in lncRNA-related studies is the careful use of databases and bioinformatics tools. Databases should be as simple as possible and always clarify the nomenclature and the manner by which RNA sequence information was obtained to become a reliable scientific resource. In the case of lncRNAs, their long sequences and the tertiary structure they usually form make them challenging to analyze biochemically and computationally. However, new in silico tools have been appearing to try to analyze the characteristics of lncRNAs under different parameters [93], so it has not been possible to establish a single replicable protocol to analyze sequences and potential targets. Moreover, some other computational tools for lncRNAs are no longer available, while others recognize themselves as still having certain inconsistencies; thus, each study has become a researcher-suitable analysis. To the extent possible, authors should consider mainly the use of tools with reliable algorithms and with a more consistent base of previous publications.

## 11. Concluding Remarks

Late diagnosis, metastasis development, and drug resistance seem to be the major limitations for a better prognosis in GC. However, RNA-seq platforms have provided valuable insight into new RNA transcripts and intracellular pathways involved in the development of GC that could provide some alternatives in the treatment and diagnosis of this malignancy.

Gastric cancer tissues and cells are usually characterized by a higher gene and protein expression of members from the PI3K/AKT/mTOR pathway or by increased post-translational activation of its proteins (mainly by phosphorylation). This can be the result of some deregulations, such as (1) genetic alterations including activating mutations in oncogenes or deletions in tumor suppressor genes, either in only one gene or in a group of genes within this pathway, and (2) epigenetic alterations, such as changes in methylation patterns of genes, histone modifications, or deregulation in the expression of non-coding RNAs, which includes lncRNAs. Both genetic and epigenetic alterations lead to an increased expression/activation of oncoproteins or reduced expression/activation of tumor-suppressor proteins of the PI3K/AKT/mTOR pathway, which leads to the development of pro-tumorigenic features, including greater cell proliferation, cell survival, migration, invasion, and chemoresistance, among others.

This review described those lncRNAs that directly or indirectly induce the upregulation of the PI3K/AKT/mTOR pathway in GC models and, therefore, could regulate the development of progression, aggressiveness, and chemoresistance features in this neoplasm. A summary of lncRNAs and their putative targets within the PI3K/AKT/mTOR pathway is shown in Figure 3.

Unfortunately, many of the articles cited in this review did not analyze in depth nor propose the molecular processes by which these lncRNAs could regulate target gene/proteins within the PI3K/AKT/mTOR pathway to generate the malignant phenotype in GC and only performed descriptive analyses regarding expression and potential targets of lncRNAs. Therefore, further analyses are needed to be performed in these specific studies in order to determine how the aggressive features commonly found in GC are developed. Conversely, other studies were able to collect more data about the molecular function that these lncRNAs would have within cells.

A special mention should be made about the potential role of lncRNAs in overcoming drug resistance in the future. These transcripts have been shown to restore sensitivity to standard chemotherapeutics and could be considered an attractive strategy for those patients that show resistance to conventional drugs or for those that could eventually develop resistance to chemical inhibitors and/or specific antibodies that try to decrease the overactivation of the PI3K/AKT/mTOR axis.

Undoubtedly, a deeper knowledge of the specific expression patterns and the molecular role of lncRNAs in cancer will be useful to determine accurate biomarkers for diagnosis or develop therapeutic strategies for GC in the future.

## Figures and Tables

**Figure 1 ijms-24-06294-f001:**
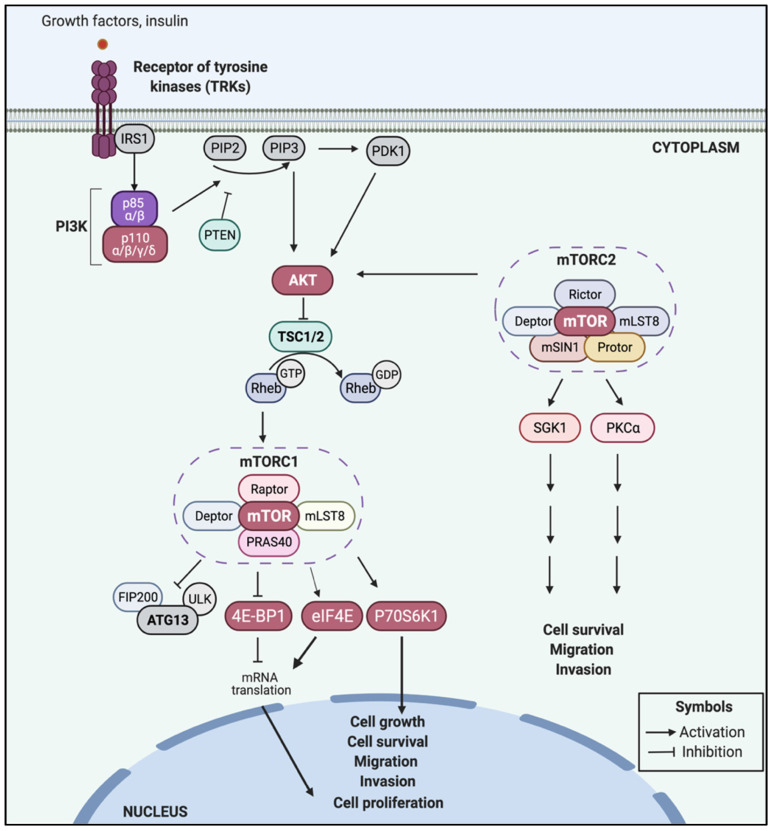
The canonical PI3K/AKT/mTOR pathway (created with Biorender).

**Figure 2 ijms-24-06294-f002:**
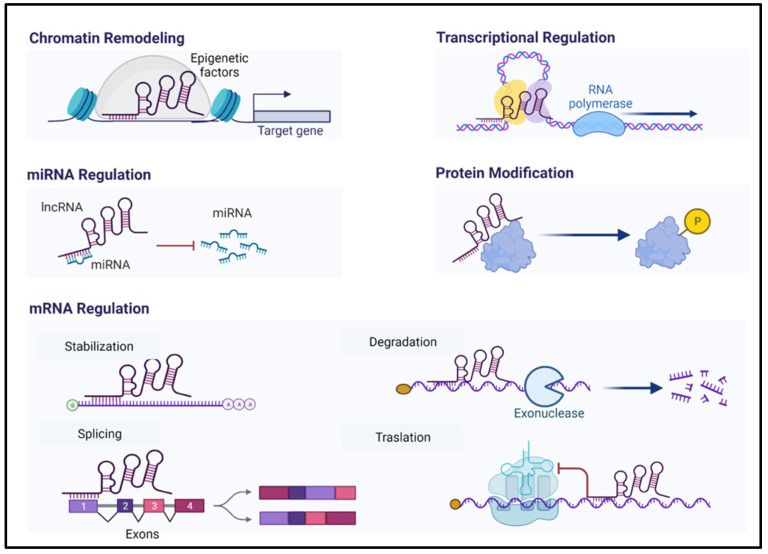
The types of epigenetic regulation exerted by lncRNAs (created with Biorender).

**Figure 3 ijms-24-06294-f003:**
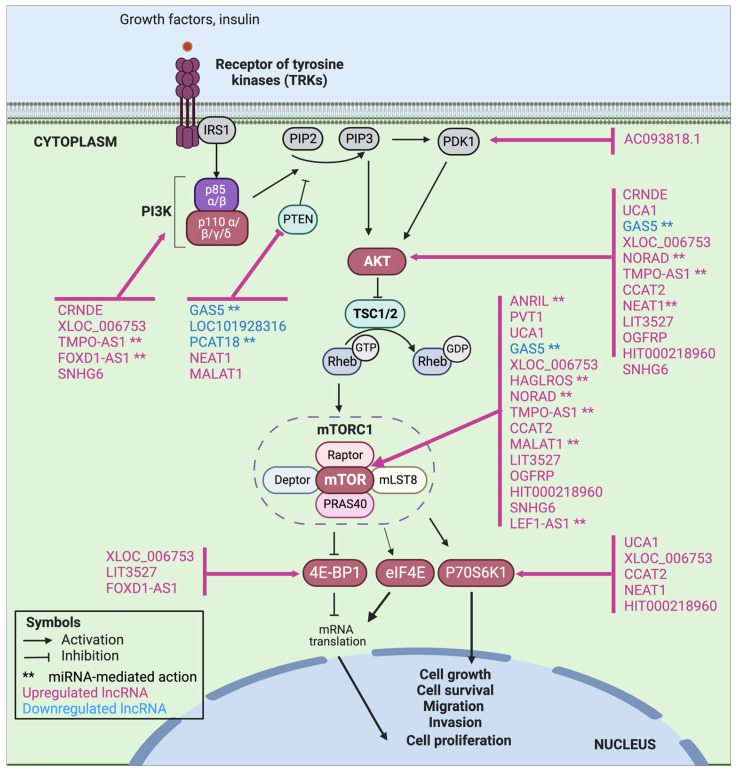
The PI3K/AKT/mTOR pathway and its current model of regulation by lncRNAs.

**Table 1 ijms-24-06294-t001:** Summary of the expression of PI3K/AKT/mTOR pathway-related lncRNAs according to clinicopathological features in gastric cancer.

lncRNA/Refs	Expression in GC	Clinicopathological Features
ANRIL [59]	Up	ANRIL overexpression was associated with greater tumor size, advanced TNM stage, and poor prognosis (OS and DFS) in GC patients.
PVT1 [60]	Up	PVT1 overexpression was frequent in GC patients with CDDP-resistant tumors.
CRNDE [61]	Up	CRNDE overexpression was correlated with invasion depth, advanced TNM stage, greater lymph node metastasis, and shorter survival (OS) of GC patients.
UCA1 [62]	Up	UCA1 overexpression was associated with advanced TNM stage and presence of metastasis in GC patients.
GAS5 [63]	Down	GAS5 expression was inversely correlated with TNM stage of tumors in GC patients.
XLOC_006753 [64]	Up	XLOC_006753 overexpression was associated with metastasis, advanced TNM stage, tumor size, and poor prognosis (OS and PFS) in GC patients.
HAGLROS [65]	Up	HAGLROS overexpression was correlated with increased invasion depth, advanced TNM stage, and poor prognosis (OS) in GC patients.
NORAD [66]	Up	NORAD overexpression was correlated with larger tumor size (>5 cm), lymph node metastasis, higher tumor grade, and advanced TNM stage in GC patients.
LOC101928316 [67]	Down	LOC101928316 expression was inversely correlated with the differentiation degree and TNM stage of gastric tumors.
PCAT18 [68]	Down	PCAT18 expression was inversely associated with tumor size.
AC093818.1 [69]	Up	AC093818.1 overexpression was correlated with invasion, lymphatic metastasis, distal metastasis, and advance TNM stage. Additionally, AC093818.1 seems to be useful to differentiate metastatic from non-metastatic GC.
TMPO-AS1 [70]	Up	TMPO-AS1 overexpression was correlated with advanced TNM stage, lymph node metastasis, and poorer survival (OS) in GC patients.
CCAT2 [71]	Up	CCAT2 overexpression was associated with greater tumor size, presence of lymph node metastasis, advanced TNM staging, and lower survival (OS) in GC patients.
MALAT1 [72]	Up	MALAT1 was significantly associated with worse pathological stage, differentiation degree, and presence of lymph node metastasis.
HIT000218960 [73]	Up	HIT000218960 expression was inversely correlated with the response to 5-FU in GC patients.
SNHG6 [74]	Up	SNHG6 overexpression was associated with lower survival (PPS) of patients, and showed a significant predictive value for the development of GC and the death of patients. Serum SNHG6 levels could be a promising prognostic marker because it would allow monitoring of GC patients before and after therapy.

Abbreviations: GC: gastric cancer; OS: overall survival; PFS: progression-free survival; PPS: Post-progression survival; CDDP: Cisplatin; 5-FU: 5-fluorouracil.

**Table 2 ijms-24-06294-t002:** Summary of the PI3K/AKT/mTOR pathway-related lncRNAs (ordered by year of study) according to phenotypic features in cancer and the mechanism by which they induce pathway activation in GC.

lncRNA/Refs	Expression in GC	Phenotypic Features	Molecular Mechanism on the PI3K/AKT/mTOR Pathway
ANRIL [59]	Up	**P, T**	ANRIL modulates the expression of miR-99a/miR-449a by binding to PRC2, thus regulating mTOR and CDK6 pathways, thereby controlling GC cell proliferation.
PVT1 [60]	Up	**A, DR**	PVT1 induces the upregulation of MDR1, MRP, mTOR, and HIF-1A genes in GC cells. *
CRNDE [61]	Up	**P, M, I**	CRNDE induces higher expression levels of p-PI3K and p-AKT in GC cells. *
UCA1 [62]	Up	**P, A, M, I, T**	UCA1 overexpression increased the expression of AKT3, p-AKT3, p-mTOR, and P70S6K1, and inhibited the expression of EIF4E. *
GAS5 [63,75]	Down	**P, A, M, I, T**	GAS5 downregulation increases miR-222 levels, thereby inducing PTEN repression and the subsequent overexpression of p-AKT and p-mTOR. Similarly, GAS5 downregulation increases miR-106a-5p levels, which also induces the overexpression of p-Akt and p-mTOR. *
XLOC_006753 [64]	Up	**P, V, A, CCP, DR, M, EMT**	XLOC_006753 overexpression induces the higher expression levels of PI3K, p-AKT, p-mTOR, P70S6K1, p-P70S6K1, and p-4E-BP1. *
HAGLROS [65]	Up	**P, M, T**	HAGLROS sponges miR-100-5p to produce an increase of mTOR mRNA expression. HAGLROS also interacts with mTORC1 components to induce an inhibition of autophagy, thereby promoting proliferation and malignant phenotype of GC cells.
NORAD [66]	Up	**P, A, T**	NORAD sponges miR-214 to finally produce an increase of p-AKT and p-mTOR levels.
LOC101928316 [67,76]	Down	**P, M, I, DR, T**	LOC101928316 downregulation induces repression of PTEN protein and, therefore, higher expression of PI3K, p-AKT, mTOR, and p-mTOR in vitro, and higher expression of p-AKT and p-mTOR in vivo. *
PCAT18 [68]	Down	**V, A, CCP, T**	PCAT18 downregulation increases miR-107 levels, which represses PTEN to activate the Akt/mTOR pathway via producing overexpression of p-PI3K and p-AKT.
AC093818.1 [69]	Up	**M, I**	In DNA, the AC093818.1 sequence overlaps on the promotor sequence of PDK1 gene; thus, this lncRNA could induce a higher expression of PDK1. In addition, AC093818.1 binds transcription factors STAT3 and SP1 to also induce a higher expression of PDK1. Therefore, AC093818.1 overexpression finally increases the levels of PDK1 and its downstream targets p-AKT1 and p-mTOR in GC cells.
TMPO-AS1 [70]	Up	**P, M, ANG**	TMPO-AS1 overexpression decreases miR-126-5p levels, which increases BRCC3 expression that subsequently would induce activation of the PI3K/Akt/mTOR pathway via producing overexpression of p-PI3K, p-AKT, and p-mTOR. *
CCAT2 [71]	Up	**P, A, CCP, ATG**	CCAT2 overexpression would induce higher levels of p-AKT, p-mTOR, and p-P70S6K1. *
NEAT1 [77]	Up	**P, A, M, I**	NEAT1 sponges miR-1294 to finally produce an increase in the AKT1 mRNA levels. Additionally, NEAT1 induces lower PTEN expression and higher levels of AKT, p-AKT, and P70S6K1. *
MALAT1 [72,78]	Up	**M, I, EMT, A, ATG, T**	The CCL21 gene expression increases MALAT1 levels, which then reduces miR-202-3p levels, inciting the SRSF1 upregulation. This last event triggers p-mTOR overexpression. * MALAT1 inhibits PTEN mRNA expression and this event could induce a greater downstream activation of the PI3K/Akt/mTOR pathway.
LIT3527 [79]	Up	**P, V, A, CD, ATG, M, MET**	LIT3527 upregulation could induce higher levels of AKT, ERK, p-mTOR, and p-4E-BP1. *
FOXD1-AS1 [80]	Up	**P, V, A, M, I, DR, T, MET**	FOXD1-AS1 increases FOXD1 expression through strengthening eIF4G- eIF4E interaction via phosphorylation of 4E-BP1. FOXD1-AS1 also sponges miR-466 to thereby produce an increase of PIK3CA expression.
OGFRP1 [77]	Up	**P, A, CCP, M, EMT, T**	OGFRP1 overexpression would induce higher levels of p-AKT and p-mTOR. *
HIT000218960 [73]	Up	**A, DR**	HIT000218960 overexpression induces increase of HMGA2 levels, which also triggers higher expression of p-AKT, p-mTOR, and p-P70S6K1. *
SNHG6 [74]	Up	**V, A, ATG, M**	SNHG6 overexpression could induce higher protein levels of PI3K, p-PI3K, AKT, p-AKT, mTOR, and p-mTOR, and lower protein levels of Beclin1 and LC3. *
LEF1-AS1 [81]	---	**A, ATG, T, MET**	LEF1-AS1 sponges miR-5100, inducing a higher DEK expression that subsequently triggers increased levels of p-mTOR. *

Abbreviations: P: Cell proliferation, V: Cell Viability; A: Apoptosis; CD: Cell death (without necessarily measuring apoptosis markers); CCP: Cell cycle progression; ATG: Autophagy; ANG: Angiogenesis; M: Migration; I: Invasion; EMT: Epithelial-mesenchymal transition; T: Tumorigenicity (increase of tumor size or weight in vivo); DR: Drug resistance; MET: Metastasis (presence of metastasis or increased migration/invasion in vivo). (*) The article did not propose a specific molecular mechanism by which activation of the PI3K/AKT/mTOR pathway would be carried out. (---) The expression levels of the specific lncRNA in GC tissues were not reported in the article.

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
