# Peer review of "Long Non-Coding RNAs (lncRNAs) as Regulators of the PI3K/AKT/mTOR Pathway in Gastric Carcinoma"

_ijms, 2023, doi:10.3390/ijms24076294_

Round 1

Reviewer 1 Report

Long non-coding RNAs (lncRNAs) as regulators of the PI3K/AKT/mTOR pathway in gastric carcinoma

 A brief summary

In this review article, the authors have first discussed the role of PI3K/AKT/mTOR pathway in gastric cancer and then reported on the regulatory role of long non-coding RNAs. Numbers of the articles used to write this article are suitable and considered to be the strength of the work.

 Considering the important role of PI3K/AKT/mTOR signaling pathway in cancer, the authors have fully explained about this pathway and also fully explained about its role in gastric cancer. Due to the fact that recently the role of lncRNAs in the regulation of vital cell processes has been identified, both these and their regulatory role in the PI3K/AKT/mTOR signaling pathway have been discussed.

General concept comment :

The major points that should be adressed are :

The article is very well written and apart from one point, there is no other point in it.

 The references used in this article are all related to the article.

Specific comments (might be redundant with the general comments for some points (

 1. The contents in lines 58 to 61 are not referenced, please add them.

Author Response

Dear Revisor,

Here are the responses to your valuable comments point-by-point 

General concept comment :
The major points that should be addressed are :
The article is very well written and apart from one point, there is no other point in it.
The references used in this article are all related to the article.

Response: Thank you very much

Specific comments (might be redundant with the general comments for some points.

 1. The contents in lines 58 to 61 are not referenced, please add them.

Response: Thanks for your comments. Topical references have been added in this paragraph.

Reviewer 2 Report

The authors present a review describing the long non-coding RNAs that have been described as regulators of the PI3K/AKT/mTOR signaling pathway in gastric carcinoma. Although there are some reviews about lncRNAs in this cancer, I believe this is the first one that focuses on the oncogenic PI3K pathway.

I think the review is correct and provides important and updated references for the field.

As minor points and as a recommendation: 

-        In the paragraph where they talk about “lncRNAs as potential diagnostic, prognostic and therapeutic markers in GC”:  If the authors decide to include this section, I think they should include here what lncRNAs have shown to be potentially useful in the diagnosis and prognosis of GC and if any of them has been detected in plasma/serum. I feel that currently this paragraph talks in general about the use of lncRNAs as potential diagnostic and prognostic markers and therapeutics targets but do not provide information about GC and expand the knowledge in this field.

-        I really like figure 3. I think it is very useful for the reader. Just as a curiosity, is there any reason why there are more upregulated that downregulated lncRNAs? In case there is one, I think it would be very interesting to include this information in the text. 

Author Response

Dear Revisor,

Here are the responses to your valuable comments point-by-point 

The authors present a review describing the long non-coding RNAs that have been described as regulators of the PI3K/AKT/mTOR signaling pathway in gastric carcinoma. Although there are some reviews about lncRNAs in this cancer, I believe this is the first one that focuses on the oncogenic PI3K pathway.

I think the review is correct and provides important and updated references for the field.

Response: Thank you very much for your comments.

As minor points and as a recommendation: 

-        In the paragraph where they talk about “lncRNAs as potential diagnostic, prognostic and therapeutic markers in GC”:  If the authors decide to include this section, I think they should include here what lncRNAs have shown to be potentially useful in the diagnosis and prognosis of GC and if any of them has been detected in plasma/serum. I feel that currently this paragraph talks in general about the use of lncRNAs as potential diagnostic and prognostic markers and therapeutics targets but do not provide information about GC and expand the knowledge in this field.

Response: Thank you very much for this comment. We have edited the initial paragraphs of this part in order to provide more understanding of the general idea of this section. You are right when you talk about we are describing the general role of lncRNAs in diagnosis and prognosis markers; however, we modified this section to give some examples of lncRNAs used as biomarkers (including references), as you asked.

-        I really like figure 3. I think it is very useful for the reader. Just as a curiosity, is there any reason why there are more upregulated that downregulated lncRNAs? In case there is one, I think it would be very interesting to include this information in the text. 

Response: Thank you very much for this comment. In general, this phenomenon is not dependent on the type of technology used in the studies nor the analyses used to obtain the results. In general, the number of upregulated lncRNAs could be similar to downregulated lncRNAs.
However, we dare to say that this matter is more oriented to what is easier to be done in the lab by the researchers. For instance, many researchers usually think that it is easier to repress an upregulated lncRNA instead effectively expressing a downregulated lncRNA. Commonly, the first is methodologically easier than the second. 
In this regard, since this matter does not refer to a technical topic and is commonly overlooked, we do not believe that it is necessary to further describe this topic in the manuscript.